# Unified Triplet-Level Hallucination Evaluation for Large Vision-Language Models

**Junjie Wu**[*]  **Tsz Ting Chung**[*]    **Kai Chen**[*]    *{junjie.wu, ttchungac, kai.chen}@connect.ust.hk*
*The Hong Kong University of Science and Technology*

**Dit-Yan Yeung**    *dyyeung@cse.ust.hk*
*The Hong Kong University of Science and Technology*

**Project Page:** `https://kaichen1998.github.io/projects/tri-he/`

**Reviewed on OpenReview:** `https://openreview.net/forum?id=iNywrSPpvc`

## Abstract

Despite the outstanding performance in vision-language reasoning, Large Vision-Language Models (LVLMs) might generate hallucinated contents that do not exist in the given image. Most existing LVLM hallucination benchmarks are constrained to evaluate the *object-related hallucinations*. However, the potential hallucination on the relations between two objects, *i.e.*, *relation hallucination*, still lacks investigation. To remedy that, we design a unified framework to measure the object and relation hallucination in LVLMs simultaneously. The core idea of our framework is to evaluate hallucinations via (object, relation, object) triplets extracted from LVLMs' responses, making it easily generalizable to different vision-language tasks. Based on our framework, we further introduce **Tri-HE**, a novel **Tri**plet-level **H**allucination **E**valuation benchmark which can be used to study both object and relation hallucination at the same time. With comprehensive evaluations on Tri-HE, we observe that the relation hallucination issue is even more serious than object hallucination among existing LVLMs, highlighting a previously neglected problem towards reliable LVLMs. Moreover, based on our findings, we design a simple training-free approach that effectively mitigates hallucinations for LVLMs. Our dataset and code for the reproduction of our experiments are available publicly at `https://github.com/wujunjie1998/Tri-HE`.

## 1 Introduction

Large Vision-Language Models (LVLMs) (Dai et al., 2023; Liu et al., 2023b; Chen et al., 2024a; Cai et al., 2024) have attracted significant attention. Despite the superior performances, existing works primarily focus on enhancing the *helpfulness* of LVLMs without careful consideration of the *reliability* of responses generated by LVLMs. However, it has already been observed by recent literature that LVLMs suffer from severe hallucination (Li et al., 2023e; Wang et al., 2023b;c; Guan et al., 2024; Chen et al., 2024b), *i.e.*, *LVLMs might generate contents that do not exist in the given image*, probably due to insufficient training during visual instruction tuning. A typical example is provided in Figure 1a, where the LLaVA (Liu et al., 2023b) model considers the location to be busy, simply because LLaVA recognizes that it is a train station with several people existing but without reasoning about their relationships.

With the prevalence of LVLMs, enormous works have started to explore the evaluation and analysis of LVLM hallucination. However, two problems are observed: 1) **Hallucination category:** most existing works focus on *object-related hallucination* (Li et al., 2023e; Wang et al., 2023b; Chen et al., 2024c) (*i.e.*, LVLM describing an object not existing in the given image) while ignoring the possibility that even when two objects

---

[*]Equal contribution.

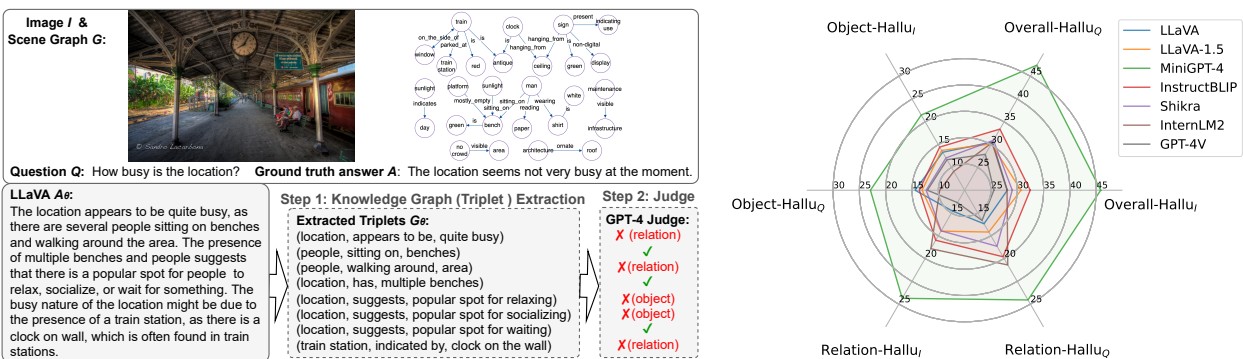

(a) **Triplet-level LVLM hallucination evaluation pipeline.**    (b) **LVLM hallucination comparison.**

Figure 1: **Overview of the unified hallucination evaluation pipeline of Tri-HE**. (a) With the provision of images, scene graphs, and questions, knowledge graphs (*i.e.*, triplets) are extracted from LVLM responses, which are then judged by an LLM (GPT-4 here). (b) The radar plot showcases the evaluation results among different LVLMs (lower values demonstrate fewer hallucinations).

are successfully recognized, LVLMs might still mess up with their relationships when conducting reasoning over these objects(Gou et al., 2025a; Yu et al., 2025). As illustrated in the example in Figure 1a, LLaVA successfully recognizes the "*people*" and the train station "*area*", yet predicts their relation to be "*walking around*" that cannot be directly obtained from the given image. Therefore, a unified definition and taxonomy is necessary to integrate different kinds of LVLM hallucination.

2) **Hallucination discrimination:** To evaluate how severe LVLMs hallucinate objects and their relationships within given images, prior works generally use either self-discrimination methods (*e.g.*, Yes/No questions) (Li et al., 2023e; Wang et al., 2023b; Guan et al., 2024; Wu et al., 2024b) or template-driven discrimination approaches (*e.g.*, "*What is the relation with A and B?*") such as Reefknot (Zheng et al., 2024). However, such methods inherently constrain LVLMs to generate short answers like "*Yes/No*" or "*A has {} relation to B*". Given that LVLMs have varying capabilities to produce brief responses due to differences in pre-training datasets, this could introduce biased evaluation results (Chen et al., 2023b; Liu et al., 2024b). For instance, Li et al. (2023e) have shown that InstructBLIP (Dai et al., 2023) tends to produce shorter outputs compared to other LVLMs, thus inflating its performance in answering the above type of questions and leads to hallucination evaluation bias. Moreover, these benchmarks require transforming general vision-language tasks into specific formats like "*Yes/No*", limiting their applicability. Therefore, we raise the following research question: **Can we develop a unified and unbiased evaluation framework capable of evaluating various types of hallucinations in LVLM responses across diverse tasks?**

To this end, we first propose a unified framework to simultaneously measure object and relation hallucinations in LVLM responses (§3). Specifically, our framework extracts knowledge graphs represented as triplets from LVLM-generated responses and then employs external evaluators to compare these triplets against the corresponding scene graphs from the input images. Consequently, our method facilitates hallucination evaluation for responses across diverse vision-language tasks, independent of the specific question formats. Leveraging this unified framework, we further introduce the **Tri-HE**, a novel benchmark for **Tri**plet-level **H**allucination **E**valuation, designed explicitly to assess both object and relation hallucinations (§4). Our experimental findings presented in §5 and Figure 1b confirm that relation hallucination poses a significant challenge for both closed-source and open-source LVLMs, often surpassing object hallucination in severity. By systematically comparing LVLMs' performance, we identify key insights that could potentially reduce hallucination rates (§5.2). Furthermore, our proposed triplet-level hallucination judge, powered by LLMs, demonstrates impressive alignment with human judgments (Table 3). Motivated by these observations, we incorporate explicit triplet descriptions into LVLM prompts and introduce a straightforward yet effective training-free method to mitigate hallucinations (§5.4).

Our primary contributions are summarized in the following three perspectives:

1. We propose a unified framework capable of jointly evaluating object and relation hallucinations in LVLM responses across diverse vision-language tasks. In particular, our triplet-level evaluation offers a finer-grained, more accurate assessment compared to existing methods.

2. Building upon this framework, we introduce **Tri-HE**, a novel triplet-level fine-grained hallucination evaluation benchmark tailored specifically for LVLMs.

3. We propose a simple yet highly effective training-free hallucination mitigation approach that surpasses the open-source LVLM competitors.

## 2 Related Work

### 2.1 Large Vision-Language Models (LVLMs)

The powerful capability exhibited by Large Language Models (LLMs) has facilitated the extension of LLMs towards the multi-modal domain. LLMs are empowered to understand and reason about both images and text by aligning representations from visual encoders to pre-trained language models, followed by visual instruction tuning. LLaVA (Liu et al., 2023a;b) proposes to use a simple projection layer to integrate the visual representations into textual encoders, which is further enhanced in Shikra (Chen et al., 2023d) by incorporating referential dialogue tasks. Instead, BLIP (Li et al., 2023a) proposes the Q-Former architecture to extract useful information from the visual representations, which is also used by MiniGPT-4 (Zhu et al., 2023) and InstructBLIP (Dai et al., 2023). InternLM (Dong et al., 2024) aligns with more diverse instruction data with the conditional online reinforcement learning from human feedback (RLHF) strategy, while MoCLE (Gou et al., 2023) further introduces the Mixture-of-Experts architecture into LVLMs to deal with the data conflict during instruction tuning. Although powerful, existing works primarily focus on improving the helpfulness and robustness Gou et al. (2025b), without a thorough analysis of the reliability of LVLMs.

### 2.2 Hallucination Evaluation in LVLMs

With the prevalence of LVLMs, a growing number of studies have been conducted on their hallucination issues (Chen et al., 2024b;d; Han et al., 2024; Huang et al., 2023; Li et al., 2023b; Wang et al., 2023b; Guan et al., 2024; Yue et al., 2024). Previous hallucination evaluation works can be categorized into two groups: 1) solely evaluating object hallucinations or do not distinguish different hallucinations (Zhao et al., 2023; Li et al., 2023c; Wang et al., 2023b; Chen et al., 2024c), which neglects other hallucination types like relation hallucination and is thus not comprehensive. The other type of works use "yes/no" questions to evaluate LVLM's relation/object hallucinations (Li et al., 2023e; Wang et al., 2023a; Guan et al., 2024; Wu et al., 2024b). However, these benchmarks require transforming general vision-language tasks into "yes/no" formats, limiting their applicability. Also, different LVLMs may have different ability in answering such"yes/no" questions since they are pre-trained on different data, which may bias the evaluation results. To remedy this research gap, our paper proposes a triplet-level evaluation framework that can provide fine-grained object and relation hallucinations for responses to any vision-language tasks, with an evaluation benchmark Tri-HE that incorporates questions requiring more complicated commonsense reasoning.

It is noteworthy that a concurrent benchmark, Reefknot (Zheng et al., 2024), similarly assesses relation hallucinations at the triplet level. However, Reefknot exhibits several limitations compared to Tri-HE. First, Reefknot constructs VQA questions based on a simple template, "*What is the relation between A and B?*", restricting both the variety of vision-language tasks that can be evaluated and the length of LVLM-generated responses, potentially introducing evaluation biases, similarly with "yes/no" questions for evaluating object hallucination. In contrast, our framework is flexible enough to be applied to various vision-language tasks. Moreover, since the questions in Tri-HE are generated by GPT-4V, it can cover a wider range of relation types compared to template-based questions, thus providing more comprehensive evaluation results. Second, Reefknot relies solely on a single entailment-based hallucination discriminator, whereas Tri-HE leverages powerful LLM-based discriminators capable of accurately and simultaneously identifying both object and relation hallucinations, leading to more comprehensive hallucination evaluation results.

# 3 Unified Hallucination Evaluation Framework Formulation

Inspired by the relation extraction (Xiaoyan et al., 2023) task, in this section, we propose a unified framework to evaluate both object and relation hallucinations via the object-relation triplets (*i.e.*, $(\text{Object}_1, \text{Relation}, \text{Object}_2)$). Here the objects and relations can either be a word or a phrase with attributes. We start by defining object and relation hallucinations via triplets in §3.1, based on which, we define our evaluation metrics and pipeline in §3.2 and §3.3 separately.

## 3.1 Definitions

As illustrated in Figure 1a, we formulate our framework with the standard VQA setting (although they can be generalized to evaluate hallucinations in any vision-language tasks given available scene graph annotations, as discussed in §3.4). Specifically, considering an input image $I$, a corresponding question $Q$ associated with image $I$, its ground truth answer $A$, and the answer $A_\theta$ predicted by an LVLM $A_\theta(\cdot|Q, I)$ parameterized by $\theta$, we can first have the following definitions:

- $G = (V, E)$ as the *scene graph* of $I$, where $V$ and $E$ refer to all the objects existing in $I$ and all the possible relations among existing objects, respectively.

- $G' = (V' \subseteq V, E' \subseteq E)$ as the *knowledge graph* that includes all the required objects and relations to answer $Q$ precisely.

- $G_\theta = (V_\theta, E_\theta)$ as the *knowledge graph* extracted from $A_\theta$, where $V_\theta$ and $E_\theta$ include all the objects and all the possible relations among objects mentioned in $A_\theta$.

Note that here all graphs can be converted to a set of triplets (*i.e.*, $G = \{(v_1, e, v_2)\}$, where $v_1, v_2 \in V$ and $e \in E$). A common nightmare in previous LVLM hallucination literature lies in the ambiguous discrimination between prediction **hallucinations** and **errors** (Ji et al., 2023). To obtain unbiased hallucination evaluation results, we separate them depending on **whether or not the wrongly generated objects or relations exist in the given image $I$**. Specifically, given a triplet $(v_1, e, v_2) \in G_\theta$, we have the following definitions,

- **Object hallucination**: if $v_1 \notin V$ or $v_2 \notin V$, suggesting $A_\theta$ includes an object not within $I$. For example, the triplet (*location, suggests, popular spot for socializing*) in Figure 1a encounters an object hallucination since the object "*popular spot for socializing*" cannot be obtained from $V$.

- **Relation hallucination:** if $v_1, v_2 \in V$ yet $e \notin E$, suggesting that $A_\theta$ correctly recognizes two related objects from $I$ but pair them with a non-existing relation. For example, the triplet (*people, walking around, area*) in Figure 1a has a relation hallucination since the relation "*walking around*" cannot be obtained from $G$, despite that the objects are all in $V$.

- **Prediction error:** if $v_1, v_2 \in V$ and $e \in E$ yet $(v_1, e, v_2) \notin G$, suggesting $A_\theta$ correctly recognizes objects and relations from $I$, yet pairs in a wrong way.

## 3.2 Evaluation Metrics

With the above definition in hand, given the knowledge graph $G_\theta$ extracted from a model response $A_\theta$, we calculate the hallucination rates of $A_\theta$ as the **proportion of hallucinated triplets** in $G_\theta$. Most previous works (*e.g.*, POPE (Li et al., 2023e)) directly evaluate the hallucination rate at the object-level with respect to the total number of predicted objects, yet make their results **not comparable among LVLMs**, since different LVLMs might refer to different numbers of objects in their responses. To address this issue, we instead opt to calculate the hallucination rate at the *question-* and *image-level*. Specifically, we calculate two types of hallucination rates, including the *question-level hallucination rate* ($\text{Hallu}_Q$) and *image-level hallucination rate* ($\text{Hallu}_I$), as defined in the following,

$$\text{Hallu}_Q(\{Q\}) = \frac{1}{|\{Q\}|} \left( \sum_{Q' \in \{Q\}} \left( \frac{\# \text{ HT in } G_\theta}{\# \text{ TT in } G_\theta} \right) \right) \times 100\%, \tag{1}$$

$$\text{Hallu}_\mathbf{I}(\{I\}) = \frac{1}{|\{I\}|} \left( \sum_{I' \in \{I\}} \text{Hallu}_\mathbf{Q}(\{Q_{I'}\}) \right) \times 100\%, \tag{2}$$

where HT is Hallucinated Triplets, TT is Total Triplets, $\{Q\}$ and $\{I\}$ are the sets of questions and images that LVLMs are evaluated on, respectively, and $\{Q_{I'}\} \subseteq \{Q\}$ suggest the subsets of questions related to the image $I'$. For both metrics, lower values demonstrate fewer hallucinations. Since the total number of questions and images is maintained the same for all evaluated LVLMs, **$\text{Hallu}_\mathbf{Q}(\cdot)$ and $\text{Hallu}_\mathbf{I}(\cdot)$ are indeed comparable and unbiased**.

### 3.3 Evaluation Pipeline

With the definitions and evaluation metrics provided in §3.1 and §3.2, the remaining problems contain two folds: 1) how to extract the knowledge graph $G_\theta$ from LVLM responses $A_\theta$, and 2) how to judge whether a triplet in $G_\theta$ is hallucinated or not. The overview of our pipeline is illustrated in Figure 1a.

**Knowledge Graph Extraction.** Given an LVLM response $A_\theta$ with the corresponding question $Q$ and image $I$, we extract the knowledge graph $G_\theta$ from $A_\theta$ via prompting GPT-4. Check our prompt for knowledge graph extraction in Appendix §A.1. Afterwards, we propose two different strategies to judge whether a triplet $(v_1, e, v_2) \in G_\theta$ includes hallucination based on the ground truth answer $A$ and the image scene graph $G$, as described in the following.

**NLI Judge.** The first strategy is implemented with a natural language inference (NLI) (Reimers & Gurevych, 2019) model [1]. Specifically, given an extracted triplet, we first calculate its cosine similarity scores with all triplets in the image scene graph $G$ and only retain those ground truth (GT) triplets with similarity scores greater than 0.5 to refine the information that will be used for the NLI model. If no triplets in $G$ meet this criterion, only the top three GT triplets with the highest similarity scores will be kept, which are then taken as ground truth inputs for the NLI model to make predictions. If the NLI score between the extracted triplet and ground truth triplets is lower than 0.6, suggesting the extracted triplet cannot be induced based on GT triplets, and therefore, resulting in a hallucination.

To determine the threshold, we randomly selected question instances from 10 images and reviewed the set of filtered triplets that were returned. The similarity score threshold was adjusted to 0.5 for the most reasonable returned triplets. These triplets later concatenate together as the ground truth required for generating NLI judgments. In determining if a generated triplet was hallucinated, we further review the NLI judgment results in different thresholds, ultimately deciding on a threshold of 0.6.

**LLM Judge.** Another evaluation strategy is to leverage prompting of a powerful LLM, a widely-adopted practice in recent works for assessing LLM outputs (Zheng et al., 2023). In this work, we primarily utilize GPT-4 in LLM judge to determine whether a given extracted triplet $(v_1, e, v_2) \in G_\theta$ can be **directly obtained** or **inferred** from the image scene graph $G$. Note that:

1. We do not employ GPT-4V in LLM judge, as Li et al. (2024) have reported that the text-only GPT-4 judge is more consistent with human preferences than GPT-4V judge.

2. Frontier open-source models, such as LLaMA-3.3, can similarly deliver reliable and cost-efficient hallucination evaluation results (see detailed analysis in Table 3), which might be superior alternatives when the GPT-4 judge is unavailable.

Additionally, if a triplet $(v_1, e, v_2)$ is judged as hallucinated, we further prompt the LLM judge to clarify whether the hallucination pertains specifically to the relation $e$ or the objects $v_1, v_2$. Refer to Appendix §A.2 for the prompt of LLM judge in our experiments [2].

---

[1] https://huggingface.co/sentence-transformers/all-mpnet-base-v2
[2] For both knowledge graph extraction and LLM judge, we utilize the "gpt-4-1106-preview" model via OpenAI's API with default inference parameters.

### 3.4 Generalizability of our Framework

Although we formulate our unified triplet-level hallucination evaluation framework in the sections mentioned above primarily based on the VQA tasks, it is capable of evaluating hallucinations in LVLM responses for any vision-language task (since knowledge graph extraction is suitable for all natural-language-based responses), provided that the corresponding scene graphs for the test images are available or extracted by pre-trained expert models. This underscores the task-agnostic design of our proposed framework and highlights its strong generalization capability. We leave the detailed exploration for other vision-language tasks for future work.

## 4 Tri-HE Construction

Following the formulation in §3, in this section, we provide a detailed discussion on how to construct our benchmark Tri-HE for a unified triplet-level evaluation of both hallucinations in LVLMs.

**Image Collection.** The construction of Tri-HE begins with images from the GQA dataset (Hudson & Manning, 2019), as the scene graph annotations provided by GQA naturally fit our triplet-level hallucination evaluation formulation. Nevertheless, some scene graphs in GQA contain incomplete object relationships, omitting information necessary for accurate question answering. To mitigate this issue, we initially filter the GQA images, retaining only those whose scene graphs contain at least five object relations (edges between nodes). Subsequently, we select 300 images from the filtered images according to the following criteria:

1. Each image must contain more than two related objects.

2. Each image must be sufficiently clear to discern all visual details.

This procedure ensures a set of high-quality images suitable for subsequent dataset construction.

**VQA Question Generation** Next, since the VQA questions in the GQA dataset have already been extensively used during the pre-training of current LVLMs, we instead employ GPT-4V [3] to generate novel question-answer pairs for each image to avoid data contamination. To effectively examine both object and relation hallucinations in LVLM responses, we aim to generate questions that necessitate commonsense reasoning grounded on the provided images. Specifically, for every image, GPT-4V is prompted to generate 10 questions along with their answers [4], each requiring image-based commonsense reasoning to be answered. Furthermore, we ask GPT-4V to produce relation triplets describing the reasoning processes of answering the questions. These additional triplets can subsequently be used to enrich the original image scene graphs.

**VQA Question Verification** Following the initial generation of VQA questions, three annotators manually examine all generated questions, answers, and triplets based on the following criteria:

1. Each question must be valid and answerable using commonsense reasoning based on the input image.

2. Each answer must appropriately address the question using commonsense reasoning.

3. Each triplet must accurately describe the corresponding answer and must only contain objects visible within the image.

Questions or answers failing to meet these conditions are discarded, while invalid triplets are excluded from the respective scene graphs. To validate the annotation consistency, the annotators jointly annotate 100 common question-answer pairs, achieving a Krippendorff's alpha (Krippendorff, 2011) of 0.62, demonstrating substantial inter-annotator agreement and guaranteeing the consistency of Tri-HE annotations.

---

[3]We use the "gpt-4-vision-preview" model here, the same as in §5.2.

[4]Check §A.3 for the prompt used here.

| #Images | #Questions | #Objects | #Relations | #Questions/Image | #Triplets/SG |
|---------|-----------|----------|-----------|------------------|--------------|
| 300 | 1226 | 1723 | 618 | 4.09 | 19.10 |

Table 1: **Statistics of Tri-HE.** "SG" refers to Scene Graph.

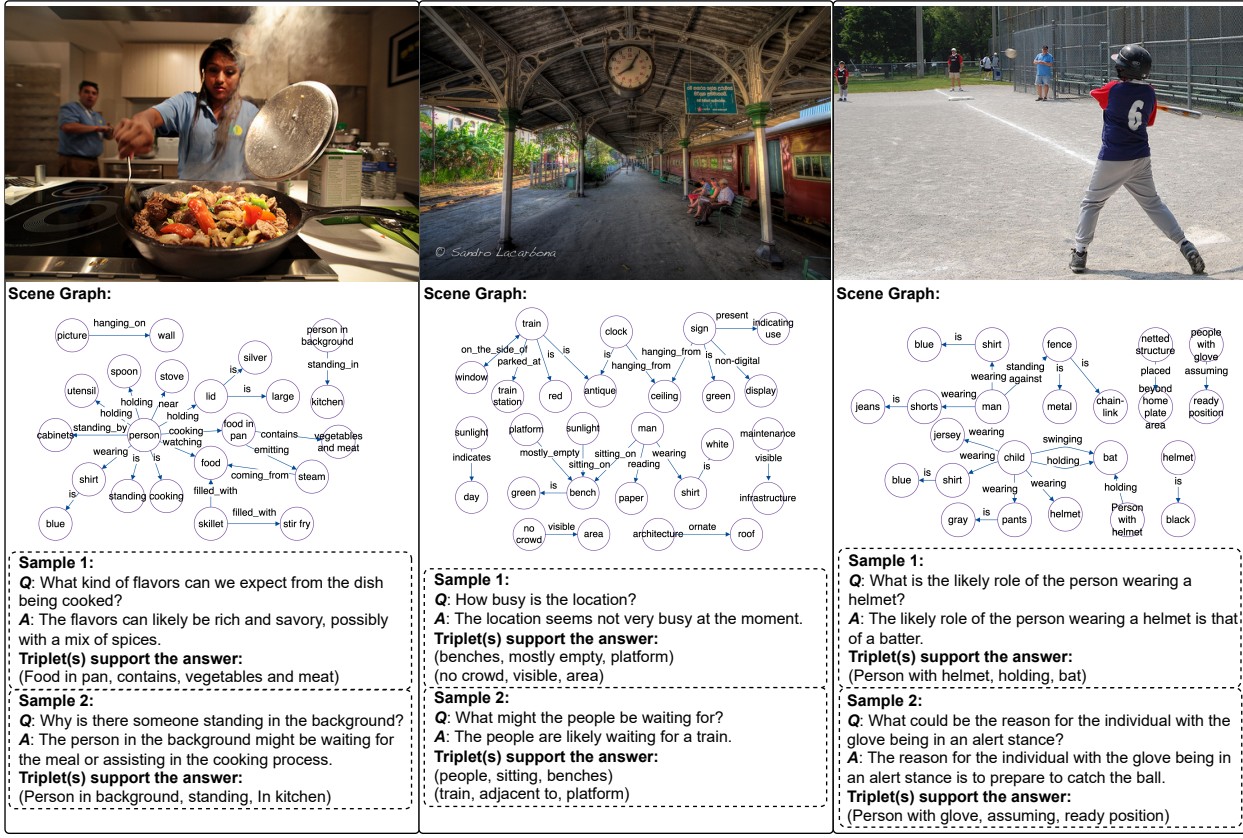

Figure 2: **Visualization of data samples in Tri-HE**. Each image is associated with a scene graph and question-answer pairs with the reasoning triplet annotations.

**Statistics.** The overall statistics for Tri-HE are summarized in Table 1. As described in Figure 2, each image in Tri-HE is linked to a scene graph and a set of question-answer pairs that require reasoning, accompanied by ground truth triplet annotations. Note that since the quality of each question in Tri-HE is manually verified, expanding its size requires significant resources and poses challenges. Nonetheless, the number of images and questions in Tri-HE is comparable to existing LVLM hallucination evaluation benchmarks such as Zhao et al. (2023) and Guan et al. (2024). Furthermore, as demonstrated in Table 3, Tri-HE has been sufficient to provide reliable hallucination evaluation results, largely thanks to the high-quality annotation procedure described above.

## 5 Evaluation Results

### 5.1 Evaluated LVLMs

We selected six open-source LVLMs for evaluation, including the LLaVA series (Liu et al., 2023a), MiniGPT-4 (Zhu et al., 2023), InstructBLIP (Dai et al., 2023), Shikra (Chen et al., 2023d), and InternLM-XComposer2 (*abbrev.*, InternLM-X2) (Cai et al., 2024). For all evaluated LVLMs, we selected the 7B variants to ensure fair comparison. Additionally, we test the recent popular Llama-3.2-Vision-Instruct (*abbrev.*, LLaMA-3.2) (MetaAI, 2024a) and use its smallest version (11B). The prompt templates and inference configurations used for LVLMs are detailed in Appendix §A.4 and §B. All experiments are conducted on two Nvidia A100 GPUs.

| Method | LLM Judge | | | | | | NLI Judge | |
| | Overall | | Object | | Relation | | Overall | |
| | $Hallu_I \downarrow$ | $Hallu_Q \downarrow$ | $Hallu_I \downarrow$ | $Hallu_Q \downarrow$ | $Hallu_I \downarrow$ | $Hallu_Q \downarrow$ | $Hallu_I \downarrow$ | $Hallu_Q \downarrow$ |
|---|---|---|---|---|---|---|---|---|
| MiniGPT-4 | 53.60 | 51.79 | 28.32 | 26.77 | 25.25 | 24.98 | 55.61 | 53.36 |
| InstructBLIP | 46.68 | 45.57 | 22.19 | 20.88 | 24.50 | 24.69 | 58.25 | 55.56 |
| LLaVA | 42.34 | 41.30 | 19.88 | 18.50 | 22.46 | 22.80 | 54.49 | 51.51 |
| Shikra | 42.20 | 41.76 | 18.55 | 17.54 | 23.65 | 24.22 | 56.46 | 53.98 |
| LLaVA-1.5 | 40.66 | 39.10 | 18.63 | **17.28** | 22.03 | 21.82 | 54.14 | 51.67 |
| LLaMA-3.2 | 40.16 | 38.95 | 22.30 | 21.08 | **17.86** | **17.87** | **48.46** | **45.64** |
| InternLM-X2 | **38.83** | **37.54** | **18.25** | 17.50 | 20.58 | 20.04 | 54.41 | 52.08 |

Table 2: **Comparison on hallucination rates among different LVLMs on Tri-HE.** The best results under each column are **boldfaced**. InternLM-X2 is short for InternLM-XComposer2 (Cai et al., 2024). Check Appendix D for evaluation results on more LVLMs.

| Method | LLaVA | LLaVA-1.5 | MiniGPT-4 | InstructBLIP | Shikra | InternLM-X2 | GPT-4V |
|---|---|---|---|---|---|---|---|
| NLI Judge(Sentence) | 0.2182 | 0.0970 | 0.3609 | 0.2596 | 0.2684 | 0.2524 | 0.2787 |
| NLI Judge(Triplet) | 0.2951 | 0.2838 | 0.2264 | 0.4259 | 0.2829 | 0.2647 | 0.4190 |
| LLM Judge(Llama-3.3 Sentence) | 0.4705 | 0.4842 | 0.3617 | 0.2520 | 0.4941 | 0.4366 | 0.4969 |
| LLM Judge(Llama-3.3 Triplet) | 0.5138 | 0.5262 | 0.4150 | 0.4798 | 0.5311 | 0.5323 | 0.5519 |
| LLM Judge(GPT-4 Sentence) | 0.6631 | 0.5409 | 0.3669 | 0.5532 | 0.5821 | 0.5998 | 0.5548 |
| LLM Judge(GPT-4 Triplet) | **0.8115** | **0.6320** | **0.4283** | **0.6235** | **0.6939** | **0.7169** | **0.7292** |

Table 3: **Pearson correlation scores** among automatic hallucination judgments and human judgments. The best results under each column are **boldfaced**. The LLM Judges utilized are specified in the brackets.

## 5.2 Main Result

**LVLM comparison.**   Table 2 compares hallucination rates of different LVLMs on our Tri-HE benchmark. As can be seen, all the evaluated LVLMs suffer from generating hallucinations with at least 38% hallucination rates. Among these LVLMs, InternLM-X2 (Cai et al., 2024) obtains the best overall performances, suggesting that its strategy to train with both text-image and textual-only instruction data simultaneously helps better align its visual encoder and LLM, and thus, reduces its hallucination rates. Moreover, compared to LLaVA (Liu et al., 2023b), Shikra (Chen et al., 2023d) has consistently lower hallucination rates, which is built upon LLaVA's structure with extra grounding capability introduced, indicating that introducing extra grounding could help LVLMs reduce hallucination. Additionally, LLaMA-3.2 achieves the lowest relation hallucination rates, suggesting that a strong textual backbone can help mitigate relation hallucination. However, it exhibits a weaker ability to accurately identify objects, impacting its object and overall hallucination rates. **Since LLaMA-3.2 does not outperform other LVLMs with even more parameters, we do not adopt it in the remaining experiments for parameter consistency.**

**Relation hallucination is more severe.**   Except for MiniGPT-4 and LLaMA-3.2, all the LVLMs generate more relation hallucinations than object hallucinations. A possible explanation is that existing LVLMs lack reasoning abilities, which makes them easily confused and mess up the relations among objects. This further suggests that focusing on object hallucination (Li et al., 2023e) is not enough for a throughout analysis of the LVLM reliability, and a unified and comprehensive study like our triplet-level evaluation is necessary.

**Evaluation pipeline.**   In addition, we observe that LLM Judge can provide clearer and more reasonable discrimination between models compared to the NLI judge. We provide a more comprehensive investigation into the differences between these two judges later in §5.3. Besides, the evaluation results under both $Hallu_I$ and $Hallu_Q$ metrics demonstrate similar trends, proving the robustness of our proposed triplet-level hallucination evaluation setting under different evaluation granularities.

|  |  | LLaVA | LLaVA-1.5 | MiniGPT-4 | InstructBLIP | Shikra | InternLM-X2 |
|---|---|---|---|---|---|---|---|
| Original | $\text{Hallu}_I \downarrow$ | 22.46 | 22.03 | 25.25 | 24.50 | 23.65 | 20.58 |
|  | $\text{Hallu}_Q \downarrow$ | 22.80 | 21.82 | 24.98 | 24.69 | 24.22 | 20.04 |
| First 20% | $\text{Hallu}_I \downarrow$ | 20.86 | 18.44 | 23.00 | 21.73 | 22.47 | 18.57 |
|  | $\text{Hallu}_Q \downarrow$ | 18.73 | 18.06 | 22.68 | 19.82 | 19.34 | 16.10 |

Table 4: **Relation hallucination rates for the top 20% frequent object pairs** of different LVLMs under the LLM Judge. **Original** refers to the results in Table 2.

**Evaluate Closed-sourced LVLMs.** In addition to evaluating open-sourced LVLMs, we further investigate the performance of closed-sourced LVLMs. Due to limited experimental resources, we specifically evaluate the GPT-4V (OpenAI, 2023) model on a subset of 25 randomly selected images from Tri-HE. Specifically, we prompt GPT-4V to obtain responses to all questions related to these selected images and compute the associated hallucination rates following the steps described in Table 2. For comparison, we also include results from open-sourced LVLMs evaluated on the same set of 25 images. As illustrated in Figure 1b, GPT-4V clearly demonstrates superior performance, surpassing all open-sourced LVLMs. Although GPT-4V exhibits slightly higher object hallucination rates compared to InternLM-X2—likely because it tends to associate additional objects not present in the image—it achieves notably lower relation hallucination rates due to its stronger reasoning capabilities, resulting in lower overall hallucination rates.

### 5.3 Analysis

**Investigating automatic hallucination judgments with human judgments.** Here, we further illustrate the effectiveness of the triplet-level evaluation setting by studying its correlation with human judgments. To conduct fine-grained hallucination analysis, previous works (Jing et al., 2023; Min et al., 2023) split a model response into sub-sentences first, on which their hallucination measurements are conducted. We regard this method as a baseline for comparison. Specifically, we sample a subset of 20 images from Tri-HE and invite human annotators to score five-point-scale hallucination rates of the responses of all the LVLMs in §5.1 (check Appendix §C for the detailed annotation guidelines). The human annotators achieve a Krippendorff's alpha score of 0.66, indicating a high inter-agreement.

Results are shown in Table 3. We find that triplet-level hallucination rates have higher correlations with human judgments with both NLI and LLM Judges, indicating that identifying hallucination on triplets can lead to a more accurate, human-preferred evaluation for model responses. Moreover, we notice that the LLM Judges achieves a higher correlation to human judgments compared to the NLI counterpart, revealing LLMs' superior abilities to find hallucinations, which is also consistent with our observation in §5.2.

**Applying Different LLMs in LLM Judge.** While GPT-4 allows the LLM Judge to produce reliable hallucination evaluations, the associated API expenses could become large when evaluating a large number of examples. To mitigate potential cost constraints, we also examine whether alternative open-source LLMs can serve effectively in LLM Judge. Specifically, we replace GPT-4 with LLaMA-3.3-70B-Instruct (*abbrev.*, Llama-3.3) (MetaAI, 2024b) and re-evaluate all examples listed in Table 3. As shown, similar to GPT-4, Llama-3.3 consistently achieves higher correlation scores at the triplet-level than at the sentence-level. Furthermore, its Pearson correlation scores with human evaluations, while significantly outperforming those obtained using NLI Judge, remain comparable to GPT-4's results for certain LVLMs. These findings suggest that open-source LLMs can serve as viable alternatives to GPT-4 in LLM Judge, providing reliable evaluation results under tight budget constraints, thereby further validating the robustness of our proposed LLM Judge.

**Investigating relation hallucination with object information.** As concluded from §5.2, existing LVLMs tend to generate both object and relation hallucinations in their replies, while the relation hallucination rates are even higher. Since different LVLMs have pairs of objects $(v_1, v_2)$ that they are familiar with (*e.g.*, high-frequency object pairs in the instruction data they are fine-tuned on) and might generate the correct relations on these objects easily, we suppose that the relation hallucination problem might mostly be located in less-frequent object pairs. To verify this assumption, we extract all object pairs for each LVLM from their respective $G_\theta$ generated from responses on Tri-HE, and rank these pairs based on their frequency. Then, we calculate each LVLM's relation hallucination rates on their most frequent object pairs.

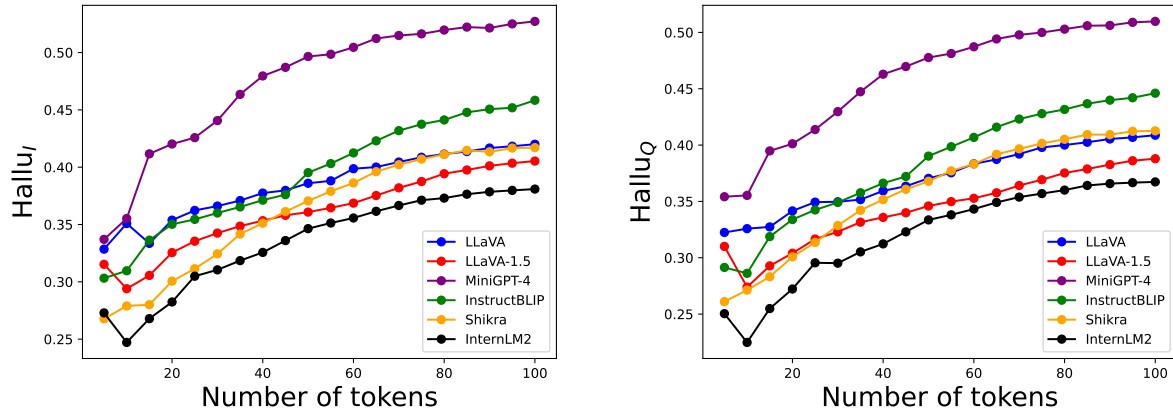

Figure 3: **Trends of the hallucination rates** of the image-level (left) and question-level (right) evaluations for different LVLMs with respect to the number of tokens in the model responses.

To obtain object pairs and their rankings from LVLM responses, suppose that for the targeted LVLM, we have its responses to all questions in Tri-HE, *i.e.*, $G_\theta$, we first extract all the object pairs $(v_1, v_2)$ from $G_\theta$. Then, for each object, we replace it with the name of its synset using WordNet to reduce the total types of objects. Afterward, we could calculate the frequency of each object pair and rank them based on their frequency. This ranking will then be used to calculate the first 20% frequent object pairs in Table 4.

As in Table 4, all the LVLMs have significantly lower relation hallucination rates on frequent object pairs they are familiar with, suggesting that they know the possible relations among objects and understand how to choose a relation appropriately for frequently occurring objects.

**Investigating hallucination rates with response length.** Previous studies on LVLM hallucination evaluation suggest that the length of model responses may influence the extent of hallucination (Li et al., 2023e; Zhou et al., 2023), as some LVLMs tend to produce shorter, safer outputs. However, directly instructing an LVLM to generate a response of a specific length is challenging. To address this, we instead truncate the responses to the first $K$ tokens and compute hallucination rates, varying $K$ to assess its impact on the results.

As shown in Figure 3, while the exact hallucination rates vary, the ranking of different LVLMs remains consistent as the number of tokens increases from 10. Overall, as fewer tokens provide insufficient data for triplet extraction, this finding supports the robustness of our proposed triplet-level evaluation across LVLMs with varying response lengths.

### 5.4 Hallucination Mitigation

After demonstrating that LVLMs exhibit significant hallucination problems, we further explore potential approaches to reduce both object and relation hallucinations. Prior works (Jing et al., 2023; Zhou et al., 2023; Li et al., 2023c; Gou et al., 2024) have suggested that *modality misalignment* might be a primary cause behind LVLM hallucinations. Motivated by this claim, we propose a training-free method to mitigate hallucinations by improving modality alignment within LVLMs.

**Method.** Specifically, we propose a two-step strategy. Given an image and its corresponding question, we first prompt the evaluated LVLM to generate a description of the image guided by the given question (**General Description** in Figure 4). Next, we prompt the same LVLM (in a new version without the image memory) using this generated description to answer the question. Through this approach, we effectively leverage the strong instruction-following capability intrinsic to the LVLM's LLM backbone, instead of requiring the LVLM to simultaneously comprehend the image and answer the question, thereby reducing hallucinations caused by modality misalignment. Moreover, as indicated in §5.3, triplet-level evaluation is more effective than sentence-level evaluation in assessing hallucinations. Hence, we further explicitly guide LVLMs to concentrate

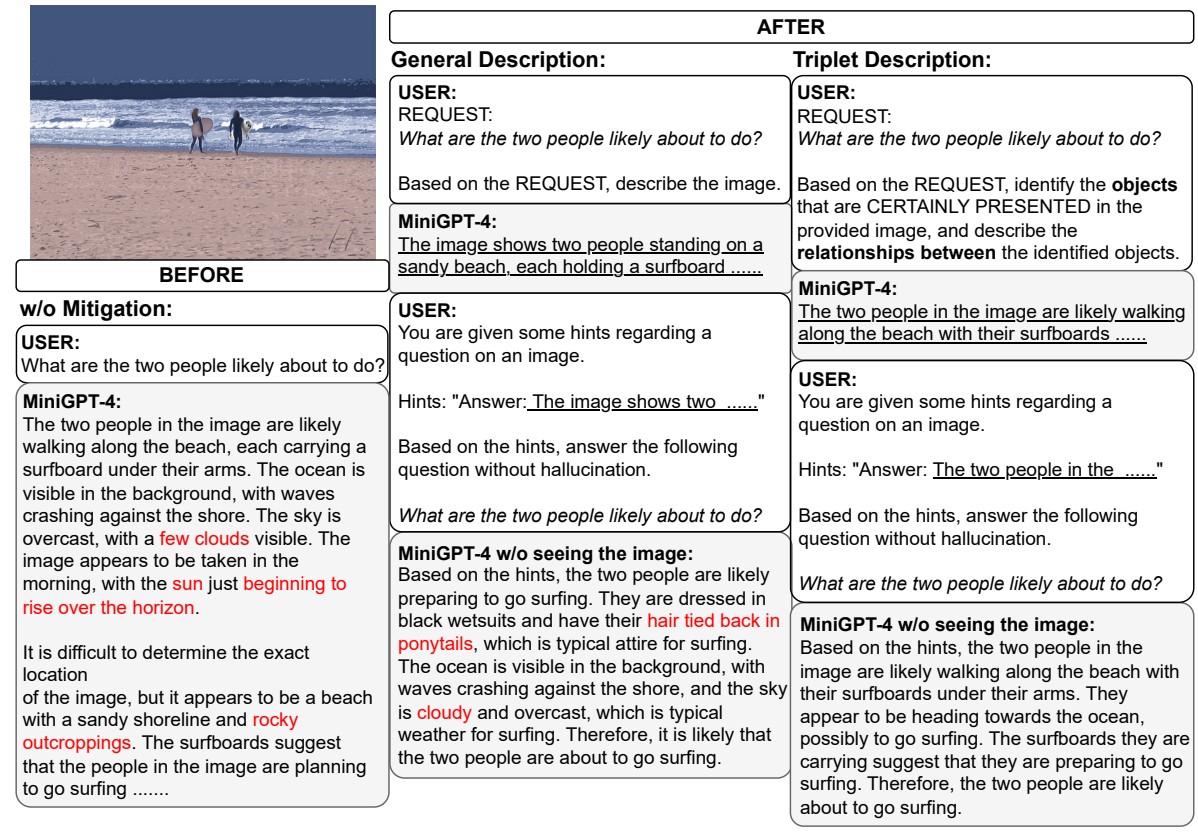

Figure 4: **An illustration demonstrating hallucination mitigation**. The three prompting strategies (w/o Mitigation, General Description, and Triplet Description) are listed from left to right. Hallucinated content is highlighted in Red and repeating contents are marked with *italic* and underline.

|  | Mitigation | LLM Judge | | NLI Judge | |
|---|---|---|---|---|---|
|  |  | Hallu$_I$ ↓ | Hallu$_Q$ ↓ | Hallu$_I$ ↓ | Hallu$_Q$ ↓ |
| MiniGPT-4 | w/o Mitigation | 45.86 | 47.44 | 55.93 | 54.94 |
|  | General Description | 46.50 | 49.19 | 54.59 | 53.03 |
|  | Triplet Description | **44.14** | **42.96** | **51.19** | **47.12** |
| LLaVA-1.5 | w/o Mitigation | 30.72 | 30.17 | 53.84 | 52.06 |
|  | General Description | 28.70 | **29.80** | 51.40 | 49.80 |
|  | Triplet Description | **28.39** | 32.68 | **48.97** | **48.40** |

Table 5: **Hallucination mitigation** results. The best results under each column are **boldfaced**.

more on identifying objects and their interrelations via triplets when describing images (**Triplet Description** in Figure 4). We evaluate MiniGPT-4 and LLaVA-1.5 combined with our proposed mitigation approaches using the subset previously employed in Figure 1b. The corresponding prompts along with an example illustration are shown in Figure 4.

**Results.** As demonstrated in Table 5, both LVLMs exhibit reduced hallucination rates after applying our two-stage mitigation method, indicating that improved modality alignment effectively alleviates hallucinations. In addition, explicitly prompting LVLMs to emphasize objects and their relationships consistently yields the lowest hallucination rates across most cases, further reinforcing the findings presented in Table 3.

**Ablation Study.** We also perform an ablation study on the best-performing "Triplet Description" variant of our mitigation approach to gain deeper insights into the role of each module within our proposed method. Specifically, we compare the Triplet Description (*i.e., Triplet+Eyes-Close*) results obtained by MiniGPT-4 with two alternative setups:

| Mitigation | LLM Judge | | NLI Judge | |
|---|---|---|---|---|
| | $\text{Hallu}_\text{I} \downarrow$ | $\text{Hallu}_\text{Q} \downarrow$ | $\text{Hallu}_\text{I} \downarrow$ | $\text{Hallu}_\text{Q} \downarrow$ |
| w/o Mitigation | 45.86 | 47.44 | 55.93 | 54.94 |
| *Eyes-Close* | 46.50 | 49.19 | 54.59 | 53.03 |
| *Triplet* | 45.65 | 45.16 | 59.35 | 55.57 |
| *Triplet+ Eyes-Close* | **44.14** | **42.96** | **51.19** | **47.12** |

Table 6: **Ablation study on hallucination mitigation.** The best results under each column are **boldfaced**.

1. *Eyes-Close:* This setting is equivalent to General Description. Image access is disabled (i.e., eyes-close) while prompting LVLM to answer the question. It is designed to assess the impact of employing triplet-level descriptions.

2. *Triplet:* This setting is similar to Triplet Description but allows image accessibility. It incorporates both the original image and the generated triplet-level description simultaneously as inputs. It is designed to examine the effects of modality alignment.

The experimental results are presented in Table 6. As shown, the combined use of triplet-level description and restriction of visual input access leads to the lowest hallucination rates. These findings further validate the design choices made in our mitigation method.

| Model | Mitigation | $\text{Hallu}_\text{I} \downarrow$ | $\text{Hallu}_\text{Q} \downarrow$ |
|---|---|---|---|
| LLaVA-1.5 | w/o Mitigation | 53.84 | 52.06 |
| | LogicCheckGPT | 51.10 | 50.84 |
| | Ours (triplet+ eyes-close) | **48.97** | **48.40** |
| MiniGPT-4 | w/o Mitigation | 55.93 | 59.94 |
| | LogicCheckGPT | 52.34 | 53.04 |
| | Ours (triplet+ eyes-close) | **51.19** | **47.12** |

Table 7: **Hallucination mitigation results on LLaVA-1.5 Liu et al. (2023a) and MiniGPT-4 Zhu et al. (2023) with baseline comparison.** The best results under each column are **boldfaced**. Check more baseline comparison results in Appendix D.

**Baseline Comparison.** Current methods for mitigating hallucinations typically involve retraining, integrating external detection models, and devising decoding strategies. Compared to existing works, our approach is a plug-and-play method that neither requires costly retraining nor relies on external models. To make a more fair comparison, we conducted experiments with LogicCheckGPT Wu et al. (2024a), a training-free approach that addresses hallucinations through prompting interactions with the help of GPT-3.5. Under the cost consideration, an evaluation was conducted only with the NLI judge. The results indicate that our method outperforms LogicCheckGPT, highlighting its effectiveness in mitigating hallucinations as shown in Table 7. Qualitative comparison with LogicCheckGPT (Wu et al., 2024a) is provided in Figure 8.

## 6 Conclusion

Starting from a unified definition of hallucinations, we propose a novel triplet-level LVLM hallucination evaluation framework for both object and relation hallucinations. Then we introduce Tri-HE, a novel triplet-level LVLM hallucination evaluation benchmark, with which, we conduct a throughout analysis of the discrepancy among object and relation hallucinations. Finally, we propose a simple yet effective training-free hallucination mitigation method, which integrates our findings regarding objects and inter-object relations.

**Acknowledgments** This work has been made possible by a Research Impact Fund project (RIF R6003-21) and a General Research Fund project (GRF 16203224) funded by the Research Grants Council (RGC) of the Hong Kong Government.

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

## A   Prompts

### A.1   Prompt for triplets extraction with GPT-4

The prompt for extracting triplets in the answer generated by LVLMs is illustrated in Figure 5.

---

Given a description of the image, please extract a KG from the text and represent the KG with triples formatted with ("subject", "predicate", "object") with readability, each triplet in a line. If 'and' or 'or' exists in the input sentence, split the objects into multiple triplets. Please do not extract from uninformative sentences.

Here are some in-context examples:

### Input:
Optimus (or Tesla Bot) is a robotic humanoid under development by Tesla, Inc. It was announced at the company's Artificial Intelligence (AI) Day event on August 19, 2021. It is planned to measure 5 ft 8 in (173 cm) tall and weigh 125 lb (57 kg). It is hard to answer whether Tesla is good to invest without more information.

### KG:
("Optimus", "is", "robotic humanoid")
("Optimus", "under development by", "Tesla, Inc.")
("Optimus", "also known as", "Tesla Bot")
("Tesla, Inc.", "announced", "Optimus")
("Announcement of Optimus", "occured at", "Artificial Intelligence (AI) Day event")
("Artificial Intelligence (AI) Day event", "held on", "August 19, 2021")
("Artificial Intelligence (AI) Day event", "organized by", "Tesla, Inc.")
("Optimus", "planned to measure", "5 ft 8 in (173 cm) tall")
("Optimus", "planned to measure", "weigh 125 lb (57 kg).")
<Done>

### Input:
The image doesn't provide information about the popularity of the song. The song "Here Comes the Boom" was originally released by American rock band Nelly in 2002 for the soundtrack of the film "The Longest Yard."

KG:
("The song 'Here Comes the Boom'", "originally released by", "American rock band Nelly")
("The song 'Here Comes the Boom'", "released in", "2002")
("The song 'Here Comes the Boom'", "featured in", "soundtrack of the film 'The Longest Yard'")
("American rock band Nelly", "released", "The song 'Here Comes the Boom'")
("The Longest Yard", "had soundtrack featuring", "The song 'Here Comes the Boom'")
<Done>

Now generate the KG for the provided input text:

### Input:
{input_text}

### KG:

---

Figure 5: Prompt for triplets extraction with GPT-4.

## A.2 Prompt for LLM Judge

The prompt for our proposed LLM judge method is illustrated in Figure 6.

---

Given a list of reference triplets ("object1", "relation", "object2") extracted from the scene graph of an image, along with a list of objects observed in this image, your task is:

Task 1. Determine if a claim triplet ("object1", "relation", "object2") is directly supported by any single triplet in the reference, or can be logically inferred from multiple reference triplets and the list of objects. Follow these steps when finishing the task:

1. Answer "yes" if the claim appears in the reference.

2. Answer "yes" if the claim can be logically inferred from one or more triplets in the reference. Consider:

a. General Inferences: Assess common associations or implications.
b. Conditional Phrases: Note phrases like "could be", "might", "suggests", which allow broader inferences.
c. Equivalence of Objects: In your judgment, treat objects of the same kind as equal. For example, "woman", "man" should be considered under the general category of "person".
d. Support from Object List: If the claim is not directly supported or inferable from the triplets, assess whether the list of objects provides additional evidence to support or infer the claim.

3. Answer "no" if the claim neither directly matches any triplet in the reference nor can be reasonably inferred from the triplets and the object list.

Task 2: Error categorization.

If your answer to the previous task is "no", determine whether the not supported/inferred part in the claim is "object1" or "object2" or "relation".

Reference:
<REFERENCE>

List of Objects:
<LIST_OF_OBJECTS>

Claim:
<CLAIM>

---

Figure 6: Prompt for the LLM Judge method.

## A.3 Prompt for question generation with GPT-4V

The prompt for generating questions, answers, and corresponding triplets with GPT-4V is shown in Figure 7.

## A.4 Prompts for Evaluating LVLMs

When evaluating LVLMs on Tri-HE, the prompt we use is the question itself. Questions are fed into LVLMs along with the corresponding images.

> Generate ten questions about the given image that require an inferential answer, which is not directly observable from the image. The answer to each question can be explained by one or more (object, relation, object) triplets that appear in the scene graph of the given image. Note that the triplets should consist of objects and relations that are visible in the given image. Output the results in the format of:
>
> Generated Questions:
>
> Answers:
>
> Explanations:

Figure 7: Prompt for question generation with GPT-4V.

## B    Configurations for LVLM Evaluation

For LVLM evaluations, we directly use the default configuration settings provided in their publicly available code repositories. For instance, the configurations utilized for evaluating LLaVA models are accessible at `https://github.com/haotian-liu/LLaVA`.

## C    Human Annotation Guideline

The detailed guidelines of our human evaluation tasks are shown in Table 8. Noting that two types of inferences in the model responses are regarded as hallucinations during human annotation:

1. Unreasonable inferences (inferences that violate commonsense knowledge).

2. Inferences that are correct, yet cannot be correctly inferred from the image.

| Score | Description |
|---|---|
| 1 | 1) The text is totally hallucinated, and is irrelevant to the given input image and question. 
 or 
 2) The text is very hard to understand. |
| 2 | 1) Most of the given responses are hallucinated, yet few sentences of them (one or two) are related to the given image and question. |
| 3 | 1) Half of the sentences in the given response are hallucinated. |
| 4 | 1) Most of the sentences in the generated response are not hallucinated. |
| 5 | 1) No hallucination exists in the generated response. |

Table 8: Detailed human evaluation instructions.

## D    Additional Results

In this section, we present additional evaluation results to supplement those in Table 2 and Table 5, enabling a more comprehensive evaluation. Due to the deprecation of `gpt-4-1106-preview`—previously employed in LLM Judge (GPT-4 Triplet)—by OpenAI's API, and our limited budget for querying proprietary models, we adopt LLM Judge (Llama-3.3 Triplet) for the following experiments. Its effectiveness has been demonstrated in Table 3.

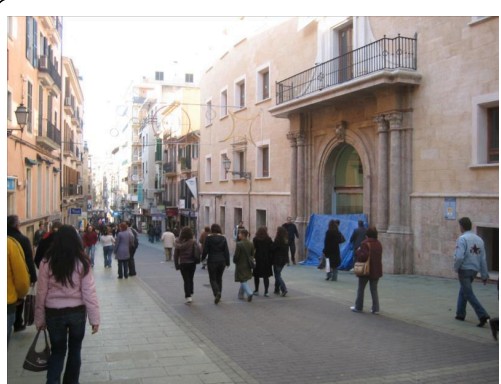

**Question:** What season could it be based on the attire of the people?

**MiniGPT-4:** Based on the attire of the people in the image, it appears to be a warm season. The majority of the people are wearing light clothing, such as shorts and t-shirts, which are suitable for warm weather. Additionally, some of the people are wearing sandals, which are also a common footwear choice for warm weather. The architecture of the buildings in the image also supports the idea of a warm season. The buildings have large windows and open balconies, which allow for good air circulation and natural ventilation, making them suitable for warm weather. The presence of greenery, such as the plants on the balconies, also suggests a warm season, as greenery tends to thrive in warm weather. Overall, the image appears to be set in a warm season.

**LogicCheckGPT [MiniGPT-4]:** The image appears to be of a group of people standing on a cobblestone street. There are several tall buildings in the background with balconies and windows. The people are dressed in warm clothing, indicating that it is likely fall or winter. Some of the people are wearing scarves and jackets, while others have on lighter sweaters or jackets. The scene is lit by streetlights, and the buildings have a white, sandy-colored facade.

**Ours (triplet+eyes-close) [MiniGPT-4]:** The people in the image are dressed in warm clothing, indicating that it is likely a winter scene. The buildings in the background are made of stone and have large windows, suggesting that they are old and possibly historic. The people are standing on the sidewalk, which is made of cobblestones and has a drainage grate in the middle. There are no cars or other vehicles in the scene, indicating that it is a pedestrian area. The overall atmosphere of the scene is quiet and still, with the people standing in a line and not interacting with each other. Based on this information, it is likely that the scene is set in the winter season.

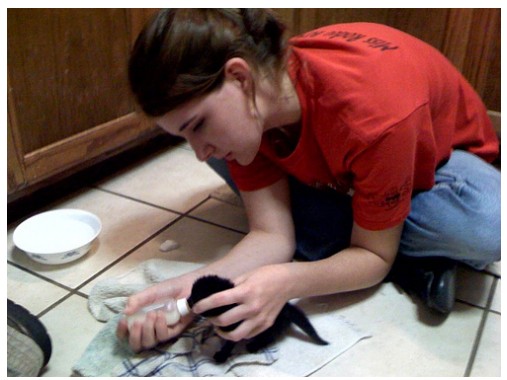

**Question:** What might the person's relationship be with the animal?

**LLaVA-1.5:** Based on the image, the person is a woman crouching down and holding a small kitten in her arms. It is likely that she has a close relationship with the kitten, possibly as its owner or caretaker. The woman's actions of holding the kitten and providing care suggest that she is familiar with the animal and is attentive to its needs. This could indicate that she is responsible for the kitten's well-being and is nurturing its growth and development.

**LogicCheckGPT [LLaVA-1.5]:** Based on the image, the woman is holding a small kitten in her arms. It is likely that she has a close relationship with the kitten, possibly as its owner. The woman's actions of holding the kitten suggest that she is familiar with the animal and is attentive to its needs. This could indicate that she is responsible for the kitten's well-being and is nurturing its growth and development.

**Ours (triplet+eyes-close) [LLaVA-1.5]:** The person's relationship with the animal is likely that of the owner or caretaker. The presence of the small black kitten lying on the floor and the person holding a bottle, which could be a baby bottle for the kitten, suggests that the person is providing care and nourishment to the kitten. Additionally, the presence of a bowl in the image supports the idea that the person is taking care of the kitten, as it could be used for feeding the kitten or providing water. This indicates a close and nurturing relationship between the person and the kitten.

Figure 8: **More illustrations demonstrating hallucination mitigation with comparison to Logic-CheckGPT**. Hallucinated content is highlighted in Red.

### D.1 Evaluating more recent LVLMs on Tri-HE

To ensure a more comprehensive evaluation, we further evaluate InternVL2_5-8B Chen et al. (2024e) and Qwen2.5-VL-7B-Instruct Team (2025) on Tri-HE. Since we use the LLM Judge (LLaMA-3.3 Triplet) in this setting—which differs from the judge used in Table 2—we also report the results of InternLM-X2 and LLaMA-3.2 under the same judge for comparison.

The results are presented in Table 9. As shown, all recent models still exhibit a certain degree of both object and relation hallucinations, demonstrating the effectiveness of Tri-HE in identifying hallucination issues in LVLMs. Notably, InternVL2_5-8B achieves the lowest hallucination rates across all metrics, suggesting that its superior pre-training data quality and the use of Random JPEG Compression as a data augmentation strategy are effective in mitigating hallucinations in model responses.

| Method | LLM Judge | | | | | |
| --- | --- | --- | --- | --- | --- | --- |
| | Overall | | Object | | Relation | |
| | $\text{Hallu}_I \downarrow$ | $\text{Hallu}_Q \downarrow$ | $\text{Hallu}_I \downarrow$ | $\text{Hallu}_Q \downarrow$ | $\text{Hallu}_I \downarrow$ | $\text{Hallu}_Q \downarrow$ |
| LLaMA-3.2 | 38.98 | 37.37 | 23.25 | 21.13 | 13.51 | 13.30 |
| Qwen2.5-VL-7B-Instruct | 38.19 | 35.92 | 26.16 | 24.09 | **11.83** | 11.38 |
| InternLM-X2 | 31.75 | 29.93 | 19.29 | 18.09 | 12.46 | 11.84 |
| InternVL2_5-8B | **29.12** | **26.99** | **17.16** | **14.95** | 12.22 | **11.13** |

Table 9: Comparison on hallucination rates among recent LVLMs on Tri-HE with LLM Judge (Llama-3.3 Triplet).

### D.2 More Hallucination Mitigation Methods

In this section, we include additional recent **training-free** methods for hallucination reduction to enable a more comprehensive comparison. Specifically, we compare our mitigation framework with a decoding-based approach, VCD (Leng et al., 2024). As shown in Table 10, although VCD is designed to address object hallucination, it exhibits more severe hallucination issues when evaluated on our reasoning-grounded question set compared to our proposed mitigation method.

In addition, reinforcement learning (RL) approaches have received growing attention for addressing hallucination. To enable a more comprehensive evaluation, we assess a **RL-trained** method, OPA-DPO (Yang et al., 2025), using our evaluation framework. The results in Table 10 demonstrate its effectiveness. It is worth noting that, since OPA-DPO requires additional training, its performance advantage over our method is expected. Nonetheless, it can potentially be integrated with our proposed approach as a foundation for further RL fine-tuning.

| Model | Mitigation | Type | $\text{Hallu}_I \downarrow$ | $\text{Hallu}_Q \downarrow$ |
| --- | --- | --- | --- | --- |
| LLaVA-1.5 | w/o Mitigation | - | 21.73 | 21.07 |
| | VCD | training-free | 28.89 | 30.64 |
| | Ours | training-free | 19.53 | 20.98 |
| | OPA-DPO | RL-trained | **14.44** | **16.82** |

Table 10: Additional baseline comparisons on hallucination mitigation with Llama3.3 Judge. The best results under each column are **boldfaced**.

## E Future works

Currently, the proposed triplet-level evaluation is primarily deployed on LVLMs, whose extension to diffusion models (Chen et al., 2023c; Gao et al., 2023; 2024a;b; Li et al., 2023d; Liu et al., 2024a; Wang et al., 2024) is feasible, while for the hallucination mitigation proposed in §5.4 can be further enhanced by utilizing

stronger vision encoder (Chen et al., 2021; 2023a; Liu et al., 2022; Zhili et al., 2023) and visual tools (*e.g.*, object detectors (Han et al., 2021; Li et al., 2022)) to better extract visual information for LLM reasoning. Furthermore, additional types of hallucination can be incorporated into our triplet-level evaluation framework, such as temporal changes of an object over time, represented in the form of (old, change, new) triplets. We plan to explore these extensions in future work.

