# OpenReview forum: "Unified Triplet-Level Hallucination Evaluation for Large Vision-Language Models"
_TMLR — Accepted by TMLR_

### Review · Reviewer_PXaj · 2025-04-15

**Summary Of Contributions:**

- **Unified Evaluation Framework**. A triplet-based method assesses object and relation hallucinations in LVLMs, using scene graphs for consistent, unbiased evaluation across models.

- **Tri-HE Benchmark**. Tri-HE offers 300 images and 1,226 verified questions to rigorously test object and relation hallucination detection in LVLMs.

- **Hallucination Mitigation Method**. A training-free technique cuts hallucinations via better modality alignment, using a two-step process with triplet emphasis for improved accuracy.

**Audience:**

No

**Broader Impact Concerns:**

* The use of GPT-4 as an LLM judge may introduce biases from its training data, potentially skewing hallucination assessments. Without transparency, this could unfairly favor certain models, affecting fairness in AI development and deployment.
* The reliance on proprietary models like GPT-4 for evaluation raises accessibility issues for researchers with limited resources. This could exacerbate inequalities in AI research, limiting diverse contributions to LVLM reliability improvements.

**Claims And Evidence:**

Yes

**Requested Changes:**

* Enlarge the Tri-HE benchmark to three times its current size, ensuring the inclusion of rare relation categories.
* Gather RL prompts and rewards training data using the expanded datasets. Execute the RL algorithm and report the results. Alternatively, for a cost-efficient approach, collect pairwise ranking data based on evaluation metrics and apply the DPO algorithm.

**Strengths And Weaknesses:**

## Strengths
- **Innovative Evaluation Framework**. The triplet-based approach uniquely evaluates both object and relation hallucinations, addressing a gap in LVLM reliability assessment.
- **Practical Mitigation Method**. A training-free technique to reduce hallucinations is introduced, making it immediately applicable without requiring additional model retraining.
- **Benchmark**. The Tri-HE dataset, with 300 carefully curated images and manually verified questions, offers a reliable resource for testing hallucination detection.
---

## Weaknesses
- **Limited Impact on Relation Hallucinations**. The paper presents a fresh perspective on relation hallucinations in VLMs, but its impact is limited by: (1) a focus on common concepts for relations, driven by Scene Graph dataset constraints, which restricts the number of relations (618 relations) and relation categories; (2) modern VLMs exhibits less hallucinations in common cases, while hallucinations are more prevalent in rare cases, yet the paper lacks results for rare cases; (3) scaling the dataset is challenging and limited due to its dependence on Scene Graphs, which itself is not scalable.
- **Limited Benchmark Size**. With only 300 images, the Tri-HE dataset may not fully represent the diversity needed for a comprehensive evaluation across various scenarios.
- **Reliance on Scene Graphs**. The evaluation framework depends on scene graphs for comparison, which could restrict applicability, especially for tasks without pre-existing annotations.
- **Limited Experimental Focus**. The experiments focus on VQA tasks, with the proposed method reducing hallucinations in VQA through captioning. However, this approach does not scale to other VLM tasks, such as captioning.
- **Limited Originality in Mitigation Approach**. The mitigation solution lacks novelty, as it mirrors approaches in prior work, essentially using a COT process to generate contextual understanding, such as caption, and include it when providing an answer.

---

### Review · Reviewer_Ev5K · 2025-04-15

**Summary Of Contributions:**

1. The authors propose a unified framework to measure object and relation hallucination in LVLMs. The framework first generates knowledge graphs from the model's response, then compares the (object, relation, object) triplets from the knowledge graph to the scene graphs of the image for hallucination evaluation.
2. Based on this framework, the authors further propose a new hallucination evaluation benchmark, Tri-HE. Tri-HE consists of images from GQA. The questions are generated by GPT-4V and filtered by human annotators. The authors evaluate several open-source LVLMs using Tri-HE. The authors conduct several analysis based on the evaluation results, such as the alignment between human and automatic judgement, different LLM's impact on evaluation results and influence of object information on hallucination.
3. The authors propose a training-free hallucination mitigation method, which first prompts the evaluated LVLM to generate a description of the image guided by the given question, then asks the LVLM to answer the given question solely based on the description.

**Audience:**

Yes

**Broader Impact Concerns:**

No concerns about the ethical implications.

**Claims And Evidence:**

Yes

**Requested Changes:**

Critical:
1. Evaluate more up-to-date LVLMs.
2. Compare the proposed methods with other training-free hallucination mitigation baselines.

Minor:
1. Using InternLM2 to represent InternLM-XComposer2 is confusing since there is also a pure language model named InternLM2.

**Strengths And Weaknesses:**

Strengths:
1. The paper is logically clear and easy to follow.

2. The paper thoroughly investigates the issue of relation hallucination in LVLMs. It designs a new evaluation framework, collects a novel benchmark, and proposes a training-free mitigation method.

3. The data annotation process is rigorous. The authors provide quantitative metrics for inter-annotator agreement in the paper.

Weaknesses:
1. Insufficient evaluation of models, with outdated versions. As a main contribution of the submission, the authors should evaluate more LVLMs on the proposed benchmark Tri-HE to provide a more comprehensive view of relation hallucination issues in LVLMs. Currently, the authors evaluate 8 open-source LVLMs and 1 closed-source LVLM on a subset. The authors could expand the evaluation to include models of more sizes. Additionally, the evaluation lacks recent and more advanced open-source models, such as the InternVL series and the Qwen2-VL/Qwen2.5-VL series. The absence of evaluations for these models weakens the reliability of claims in the paper, such as "existing LVLMs lack reasoning abilities, which makes them easily confused and mess up the relations among objects" (Section 5.2).

2. Necessity of NLI-based evaluation. Tri-HE employs two evaluation methods: NLI-based and LLM-based. However, based on Table 3, the NLI-based evaluation shows significantly lower agreement with human judgments. I doubt its reliability as an evaluation method.

3. Effectiveness of the proposed mitigation method. Many prior works have proposed training-free approaches to mitigate hallucinations in LVLMs (e.g., VCD[1], OPERA[2], etc.). The authors do not compare their method with these existing approaches.

[1] Leng S, Zhang H, Chen G, et al. Mitigating object hallucinations in large vision-language models through visual contrastive decoding[C]//Proceedings of the IEEE/CVF Conference on Computer Vision and Pattern Recognition. 2024: 13872-13882.

[2] Huang Q, Dong X, Zhang P, et al. Opera: Alleviating hallucination in multi-modal large language models via over-trust penalty and retrospection-allocation[C]//Proceedings of the IEEE/CVF Conference on Computer Vision and Pattern Recognition. 2024: 13418-13427.

---

### Review · Reviewer_mps8 · 2025-04-20

**Summary Of Contributions:**

This work makes three key contributions: first, it defines a unified triplet‑level evaluation framework that rigorously captures both object and relation hallucinations by extracting and verifying (subject, relation, object) triplets from LVLM outputs; second, it introduces Tri‑HE, a carefully curated benchmark of 300 images and over 1,200 commonsense reasoning questions (with enriched scene graphs) that enables fair, fine‑grained comparison of models via the novel HalluQ and HalluI metrics; and third, it demonstrates that powerful LLM judges (e.g., GPT‑4 or LLaMA‑3.3) align closely with human judgments for hallucination detection and leverages this insight to propose a simple yet effective training‑free mitigation strategy—two‑step prompting with focused triplet descriptions—that consistently reduces hallucination rates across multiple open‑source LVLMs.

**Audience:**

Yes

**Broader Impact Concerns:**

The rest potential issue is about ethical and privacy which not deeply discussed in this paper. For example, scene graphs (from GQA/Visual Genome) may have annotations about people or objects; if an LVLM hallucinates something about a person in an image (say, infers an occupation from clothing that’s not actually evident), is that counted? The framework would count that as a hallucination (since the occupation attribute isn’t in the scene graph). This is good – it discourages models from making stereotyped assumptions (a subtle form of hallucination). It might be worth mentioning that reducing hallucinations also helps reduce stereotypical inferences when those are not grounded in the image, which is a positive ethical outcome. On the flip side, if scene graphs have biases or omissions, a model might get penalized for a correct statement that wasn’t labeled.

**Claims And Evidence:**

Yes

**Requested Changes:**

The requested changes can be addressed according to the weakness points made in last section.

**Strengths And Weaknesses:**

# Strengths

1. This paper introduces a triplet-level hallucination evaluation framework for LVLMs, which is a notable shift from prior approaches. The proposed framework evaluates hallucinations in a unified manner, capturing both object-based and relation-based hallucinations within model outputs. It does so by extracting knowledge triplets from the model’s response and comparing them against the image’s ground-truth scene graph. This is conceptually novel: instead of judging a whole sentence or answer, the evaluation judges the factual correctness of each extracted (object1, relation, object2) triplet.

2. The Tri-HE benchmark (Triplet-level Hallucination Evaluation) built on this framework is a fine-grained testbed specifically for hallucinations. The authors highlight that this offers a finer-grained, more accurate assessment compared to existing methods. Unlike earlier benchmarks that might require task-specific transformations (e.g., forcing every query into a yes/no format), Tri-HE can be applied to diverse vision-language tasks without such restrictions, as it operates on the universal representation of triplets. This generality is the contribution – it addresses the research question the authors pose about developing a unified, unbiased evaluation across diverse tasks.

3. The paper introduces two metrics: HalluQ (question-level hallucination rate) and HalluI (image-level hallucination rate). Both metrics quantify the proportion of hallucinated triplets in model responses, but at different granularities. HalluQ is calculated per question: for each question, take the number of hallucinated triplets in the model’s answer divided by the total number of triplets in that answer, then average this fraction over all questions. HalluI is calculated per image: for each image, average the HalluQ of all questions pertaining to that image, then average over images. Essentially, HalluQ treats each QA pair equally, and HalluI treats each image equally, which helps smooth out any variance due to some images having more questions than others. Both are expressed as a percentage of hallucinations (lower is better). Another important aspect is that the evaluation distinguishes object vs relation hallucination rates (and even “prediction errors”) internally, even if HalluQ/HalluI combine them into one number. The radar chart in Figure 1b and associated discussion separate “Object-Hallu” and “Relation-Hallu” for each model, which is very informative. It appears the metric can be broken down by type: e.g., compute HalluQ but considering only triplets that are object hallucinations, versus only relation hallucinations. This flexibility is good; it allows diagnosing what kind of hallucinations a model is prone to. The paper finds that relation hallucination poses a significant challenge, often surpassing object hallucination in severity, a conclusion drawn from these metrics. That insight would have been obscured if the metric didn’t differentiate hallucination types at all.

# Weakness

1. The authors explicitly considered some biases of previous evaluations (like models giving shorter answers to game yes/no tests) and designed against them. Their approach is largely task-agnostic, requiring only that an image have a scene graph. However, one dependency is exactly that: many real-world datasets or scenarios don’t have ground-truth scene graphs readily available. This means the framework currently shines for benchmarking and analysis, but applying it beyond curated datasets might require an automated way to get image annotations (perhaps via an off-the-shelf detector or visual grounding system). The paper’s scope is a controlled evaluation, so this is not a flaw, but it’s a practical limitation to be aware of. Another point is the reliance on GPT-4 throughout: it generates data, extracts triplets, and judges hallucinations. This heavy reliance on one AI system could pose a risk if that system had consistent blind spots.

2. The authors’ approach is essentially measuring precision of the informational content (i.e., what fraction of stated facts are incorrect?). An alternative could be to also measure something akin to recall – e.g., does the model miss mentioning relevant facts? However, recall is less relevant to hallucination (it’s more about completeness than avoiding falsities), so the choice to focus on hallucination rate (precision error) is appropriate. Another alternative metric could be a severity-weighted score – perhaps not all hallucinations are equal (saying “clock on the wall” when there is none might be less severe than hallucinating a non-existent person, depending on context). But quantifying severity would introduce subjectivity, and the current metrics keep it simple and objective (a triplet is either in the image or not).

3. More experiments is needed to confirm the evaluation robustness and showing broader applicability.
  - The framework is touted as task-agnostic (it can work for any vision-language task given a scene graph). A valuable experiment would be to demonstrate this generality. For example, the authors could apply their triplet-based hallucination evaluation to an image captioning task or a visual dialogue task. In such a setup, instead of a specific question, the model generates an open-ended description. The knowledge graph extraction prompt could be adapted accordingly. If the framework works well (perhaps by using ground-truth captions or region descriptions as the “scene graph”), it would show that Tri-HE is not limited to the VQA-style setting. Even a small-scale test on a dataset like MS-COCO with ground-truth captions (converted to pseudo-triplets) could be illustrative.
  - The authors could perturb the input to see if the hallucination evaluation is robust. For example, if there are slight errors in the scene graph (simulating a less-than-perfect ground truth), does the evaluation metric degrade gracefully? Or if the question is phrased differently but asks the same thing, do models hallucinate differently? The latter could be interesting: ask two semantically identical questions phrased differently and see if the model’s hallucination rate changes. This might reveal if certain phrasings induce more imaginative answers. It’s a bit tangential to the main focus, but any findings could be useful for understanding how to query LVLMs to avoid eliciting hallucinations.

---

### Review · Reviewer_TPUu · 2025-04-20

**Summary Of Contributions:**

In this paper, the author proposed a united framework to evaluate object hallucination and relation hallucinations in LVLMs. In particular, the authors proposed a triplet evaluation to offer an accurate assessment of existing methods. The author also proposed Tri-HE benchmark that contains 1226 questions based on 300 images. Finally, the author proposed a training-free hallucination mitigation approach.

**Audience:**

Yes

**Claims And Evidence:**

Yes

**Requested Changes:**

Please see the weakness in the previous section.

**Strengths And Weaknesses:**

Strength:
1. The paper is well written and easy to understand.
2. The construction of the triplets and use of scene graph is novel and very interesting.
3. Based on their findings, a mitigation method is proposed based on iterative prompting using triplets relations.


Weakness:
1. Evaluation and comparisons are limited. The author should add more evaluation and comparisons, as a benchmark paper.
2. The size of the dataset is too small concerning. The author should enlarge the dataset and putting in more efforts in the dataset and benchmark. One of the big concerns is that the annotation of the scene graph is very difficult to scale up.
3. The author should add more evidence on the effectiveness of their triplets and compare with some baseline training-free mitigation methods. To further improve the paper, can the construction of the triplet be useful in the training-based approach?

---

### Review · Reviewer_xQk8 · 2025-04-21

**Summary Of Contributions:**

This paper introduces Tri-HE, a new benchmark for evaluating hallucination in Large Vision-Language Models (LVLMs) at the triplet level, covering both object and relation hallucinations. The proposed framework extracts (subject, relation, object) triplets from model responses and evaluates them using either natural language inference (NLI) or a large language model (LLM) judge. The authors claim this unified framework enables a fine-grained and generalizable assessment of hallucinations across tasks.

**Audience:**

Yes

**Claims And Evidence:**

Yes

**Requested Changes:**

1. Explicitly acknowledge and cite Mementos in the related work and clearly delineate how Tri-HE provides additional insights or functionality not already covered by Mementos.

2. Test a broader range of recent models: The evaluation currently covers a limited set of LVLMs, primarily those released in 2023. To better assess the generalizability and robustness of the proposed framework, I recommend including more recent state-of-the-art models . These models have made significant progress in reducing hallucinations, and benchmarking them would provide a more up-to-date and competitive comparison.

3. Expand the benchmark in both scale and scenario diversity: The current Tri-HE benchmark includes only 300 images and focuses narrowly on VQA-style prompts. To improve its utility and impact, I strongly suggest expanding the dataset in terms of both size and coverage—adding more images, a wider range of question types, and more diverse visual scenarios (e.g., indoor vs. outdoor, real-world vs. synthetic). This would better validate the general applicability of the framework and enable more robust model comparisons.

**Strengths And Weaknesses:**

Strengths

1. The paper addresses an important and timely issue in LVLM evaluation—hallucination beyond object misidentification, extending into relation-level reasoning.

2. The paper is well-writen and easy to follow

Weaknesses

1. High conceptual and methodological overlap with prior work (Mementos):  The proposed evaluation framework bears striking resemblance to the Mementos benchmark [1], which also extracts structured representations (object-behavior pairs) from multimodal model outputs, uses GPT-4 as a judge, and evaluates hallucination severity across object and relational reasoning. The key idea—extracting structured tuples and comparing them to ground-truth annotations using LLM assistance—is central in both papers.

2. No citation or discussion of Mementos: Despite the obvious similarities in task definition, evaluation pipeline (structured reasoning units judged by GPT-4), and even the conclusion that behavioral (relation) hallucinations are more problematic than object hallucinations, the current paper fails to cite or even acknowledge the Mementos benchmark. This omission significantly undermines the contribution claim and raises concerns of originality.

3. Marginal novelty: While Tri-HE targets single images and Mementos targets image sequences, this distinction is not sufficient to justify a new benchmark in the absence of novelty in methodology. Both frameworks aim to evaluate hallucination in structured formats via LLM judgment—thus, the technical advance is incremental at best.

Reference:

[1] "Mementos: A Comprehensive Benchmark for Multimodal Large Language Model Reasoning over Image Sequences" Wang et. al. https://arxiv.org/abs/2401.10529

---

### Decision · Action_Editor_iC6B · 2025-06-09

**Recommendation:** Accept with minor revision

**Audience:**

Yes

**Audience Explanation:**

The paper presents a study on hallucination in LVLMs.  It can be improved by addressing from several limitations that affect its comprehensiveness and potential impact:

- Limited Model Coverage and Recency: The evaluation includes only 9 LVLMs and does not cover the full benchmark. Notably, several recent and stronger models—such as Molmo, InternVL, and Qwen—are omitted, which limits the relevance and completeness of the analysis. In terms of methods, it is encouraged to consider RL method to reduce hallucinations, based on the evaluation metric. This would provide interesting empirical findings.

- Lack of Comparison with Prior Work: The proposed mitigation approach is not evaluated against existing training-free methods for hallucination reduction in LVLMs, such as VCD and OPERA. This omission weakens the paper’s claims regarding effectiveness and novelty.

- Narrow Scope and Limited Novelty: The work focuses on a specific type of hallucination, namely (object, relation, object) triplets. While this is a valid problem, the contribution appears incremental in the context of the broader visual hallucination literature.

**Claims And Evidence:**

Yes

**Claims Explanation:**

- Tri-HE Benchmark. 300 images and 1,226 verified questions to evaluate object and relation hallucination in LVLMs.

- Mitigation Method. A training-free technique to reduce hallucinations via better modality alignment, using a two-step process with triplet emphasis for improved accuracy.